# Speaking up behavior and cognitive bias in hand hygiene: Competences of German-speaking medical students

Stefan Bushuven[1,2,3]*, Markus Dettenkofer[2], Sonia Sippel[4], Sarah Koenig[4], Stefanie Bushuven[5], Wulf Schneider-Brachert[6]

1 Institute for Anesthesiology, Intensive Care, Emergency Medicine and Pain Therapy, Hegau-Bodensee Hospital Singen and Hegau Jugendwerk Gailingen, Healthcare Association Constance (GLKN), Radolfzell, Germany, 2 Institute for Hospital Hygiene and Infection Prevention, Healthcare Association Constance (GLKN), Radolfzell, Germany, 3 Institute for Didactics and Educational Research in Medicine, Clinic of the University Munich, LMU Munich, Munich, Germany, 4 Institute of Medical Teaching and Medical Education Research, University Hospital Wuerzburg, Wuerzburg, Germany, 5 Clinic for Orthopedics, Hand- and Trauma surgery, Hegau-Bodensee-Hospital Singen, Healthcare Association Constance (GLKN), Radolfzell, Germany, 6 Department of Infection Control and Infectious Diseases, University Hospital Regensburg, Regensburg, Germany

* S.Bushuven@gmx.de

**Data Availability Statement:** All relevant data are within the manuscript and its Supporting Information files.

## Abstract

### Introduction

Infection prevention and speaking up on errors are core qualities of health care providers. Heuristic effects (e.g. overconfidence) may impair behavior in daily routine, while speaking up can be inhibited by hierarchical barriers and medical team factors. Aim of this investigation was to determine, how medical students experience these difficulties for hand hygiene in daily routine.

### Methods

On the base of prior investigations we developed a questionnaire with 5-point Likert ordinal scaled items and free text entries. This was tested for validity and reliability (Cronbach's Alpha 0.89). Accredited German, Swiss and Austrian universities were contacted and medical students asked to participated in the anonymous online survey. Quantitative statistics used parametric and non-parametric tests and effect size calculations according to Lakens. Qualitative data was coded according to Janesick.

### Results

1042 undergraduates of 12 universities participated. All rated their capabilities in hand hygiene and feedback reception higher than those of fellow students, nurses and physicians (p<0.001). Half of the participants rating themselves to be best educated, realized that faulty hand hygiene can be of lethal effect. Findings were independent from age, sex, academic course and university. Speaking-up in case of omitted hand hygiene was rated to be done seldomly and most rare on persons of higher hierarchic levels. Qualitative results of 164

**Funding:** The author(s) received no specific funding for this work.

**Competing interests:** The authors have declared that no competing interests exist.

entries showed four main themes: 1) Education methods in hand hygiene are insufficient, 2) Hierarchy barriers impair constructive work place culture 3) Hygiene and feedback are linked to medical ethics and 4) There is no consequence for breaking hygiene rules.

## Discussion

Although partially limited by the selection bias, this study confirms the overconfidence-effects demonstrated in post-graduates in other settings and different professions. The independence from study progress suggests, that the effect occurs before start of the academic course with need for educational intervention at the very beginning. Qualitative data showed that used methods are insufficient and contradictory work place behavior in hospitals are frustrating. Even 20 years after "To err is human", work place culture still is far away from the desirable.

## Introduction

Patient safety is a core competency of all health care providers working in volatile, uncertain, complex and ambiguous working environments [1]. Medical error is a leading cause of in-hospital death and impairment [2], demanding for improvement in work place safety culture, individual habits, institutional high reliability and medical education of under- and postgraduates. Especially under-graduates (e.g. medical students) face substantial barriers and dilemma to speak up and give corrective feedback, in particular to their supervisors and educators [3, 4].

Aside from feedback skills, but noticeably linked to them, infection prevention and early treatment of health care associated infections are keystones of patient safety [5]. Especially hand hygiene plays a crucial role [5, 6]. Unfortunately, low adherence to hand hygiene protocols [6] and impaired commitment to educational lessons is challenging [7]. The causes for the reduced incentive motivation [8, 9] to participate in trainings remain unclear.

In addition to psychological and social factors like peer pressure [10] and Hawthorne's effect [11], our work group identified an overconfidence effect for infection control practices and speaking up behavior in a regional [12] and nationwide study [13]. Overconfidence is a popular [14] and robust psychological effect of flawed self-assessment. It is innate to human behavior in health, forensics and economy [15, 16] and consists of three sub-effects: a) absolute overconfidence or overestimation (to rate one-self to be better than actual measurements demonstrate), b) relative overconfidence or overplacement (the belief to be competent above the average of a population or group) and c) overprecision (the excessive belief to be true in one's own estimations) [17]. Main causes for these effects were identified as information deficits, neglect, errors of omission, incomplete feedback, self-focus and egocentrism [15] With deep knowledge about ourself, focus on ourself and overall less or inprecise information and "self-serving" assumptions about capabilities of others this leads to failing objectivity putting everybody at higher competence in comparison to others.

The condition to unintentionally rate one-self's hygiene and communication abilities to be better than the competencies of other persons is likely to lower the intrinsic motivation to attend infection control trainings [9]: "*Why should I go to a time-consuming training, if the others are more in need of it?*". This may be additionally aggravated by the diffusion of

responsibility ("*Why should I clean my hands, when nobody does*?") [18] and corrupted model learning [19] may then further impair hand hygiene behavior.

This combination of unrealistic self-assessment, consequently reduced learning-motivation paired with de facto insufficient hygiene and feedback competences [13, 20–22] and behavior in providing feedback [23] impairs patient safety and interprofessional collaboration.

Based on two published studies of our work group demonstrating overconfidence to be present already in post-graduates, we hypothesized, that the effect depends on the individual education level.

The aim of this international cross-sectional study was to evaluate the following hypotheses:

1. Overestimation and overplacement for hand hygiene and feedback skills are present in medical students

2. Overconfidence in hygiene, communication skills and subjective speaking-up behavior depend on semester, age, gender, preexisting vocational education and university curriculum

Additionally, quantitative and qualitative survey assessments were accomplished to get more insight into under-graduates' perceptions and attitudes towards patient safety culture in infection prevention.

To do so, we investigated medical students' experiences of speaking-up behavior of (other) under-graduates, physicians (of different educational and hierarchical level) and nurses in Germany, Switzerland and Austria.

## Material and methods

From May 2018 to April 2019 we used an anonymous cross-sectional online survey previously tested on 51 medical students from the personal networks of the investigators for content and structural validity as well as reliability (Cronbach's Alpha 0.89). The survey was designed by an interprofessional team of medical educators, microbiologists, hygiene experts and psychologists basing on the SATIS-1 questionnaire [12] with comparable reliability.

According to the Ethics Committee Stuttgart (Physicians Association Baden-Wurttemberg) under consideration of the Declaration of Helsinki no ethical approval of this voluntary and openly distributed study using anonymous data only is needed (Decision on 20th of September 2018, signed by Prof. Dr. Gerd Mikus). Participants were informed by the first page of the study about objectives and IP-Address anonymization by survey provider (Eunuvo GmbH, Zuerich)

The survey consisted of 36 German items (see Table 1). Items 1–9 evaluated demographic data and prior apprenticeships in a medical profession. Items 10–26 assessed the subjective personal and others' competencies in infection prevention and communication using a 5-point Likert scale with no (0 points), poor [1], partial [2], fair [3] and complete [4] agreement. Items 27 and 28 were estimations of possible hazards of an omitted hand disinfection. In items 29–31, participants assessed identification of WHO indications for hand hygiene, adherence to hygiene protocols and the likelihood of corrective feedback to and by patients, relatives and non-medical (cleaners) and medical staff of different hierarchical levels (students, interns, consultants, heads of departments). Items 32 to 35 were questions about suggestions for medical education in hand hygiene and rankings of the importance of patient safety issues. Item 36 comprised a section for further written comments (character count unlimited) on infection prevention and feedback. To limit selection bias and to provide the possibility of a completer-non-completer comparison [24] main questions were located at the beginning of the survey.

**Table 1. Survey items.** Items were presented in German language.

| No. | Variable |
|---|---|
| 1 | **What is your current study semester?** |
| | (In welchem Studiensemester befinden Sie sich gerade?) |
| 2 | **What is your gender?** |
| | (Welches Geschlecht haben Sie?) |
| 3 | **What is your age?** |
| | (Wie alt sind Sie?) |
| 4 | **Where do you currently study?** |
| | (Wo studieren Sie aktuell?) |
| | # = University No. |
| 5 | **Did you ever change your university?** |
| | (Haben Sie Ihren Studienort schon einmal gewechselt?) |
| 6 | **Did you finish a medical education as a health care worker?** |
| | (Haben Sie eine abgeschlossene Ausbildung in einem medizinischen Beruf?) |
| 7 | **What are you studying?**[*] |
| | (Welchen Studiengang absolvieren Sie derzeit?) |
| 8 | **In what semester did you experience your first teaching in hand hygiene?** |
| | (in welchem Semester erfolgte Ihre erste Unterweisung in der Händehygiene?). |
| 9 | **With regard to the lesson plans, In what semester is the first first teaching in hand hygiene accomplished?** |
| | (In welchem Semester erfolgt/erfolgte gemäß Lehrplan die erste Lehrveranstaltung zum Thema Krankenhaushygiene Ihrer Fakultät?) |
| 10 | **I conduct hand hygiene if indicated in a situation** |
| | (Ich führe die hygienische Händedesinfektion situationsgerecht durch.) |
| 11 | **I identify the indications for hand hygiene** |
| | (Ich erkenne die Indikationen für eine hygienische Händedesinfektion) |
| 12 | **I correctly choose the indicated disinfection agent for hand hygiene** |
| | (Ich wähle das benötigte Händedesinfektionsmittel situationsgerecht aus.) |
| 13 | **I identify mistakes in the conduction of hand hygiene in other persons** |
| | (Ich erkenne Fehler in der Durchführung der hygienischen Händedesinfektion bei anderen Personen.) |
| 14 | **I correct others readily, if I perceive a mistake in hand hygiene** |
| | (Ich greife korrigierend ein, wenn ich einen Fehler in der Durchführung der hygienischen Händedesinfektion bemerke.) |
| 15 | **Depending on the situation I accept feedback if others correct me for an error in hand hygiene** |
| | (Ich nehme situationsgerecht Rückmeldungen an, wenn ich durch eine andere Person auf einen Fehler in der hygienischen Händedesinfektion hingewiesen werde) |
| 16 | **Fellow students conduct hand hygiene if indicated in a situation** |
| | (Meine Kommilitonen setzen die hygienische Händedesinfektion situationsgerecht um) |
| 17 | **Fellow students identify the indications for hand hygiene** |
| | (Meine Kommilitonen wissen die jeweiligen Indikationen für eine hygienische Händedesinfektion) |
| 18 | **Fellow students correctly choose the indicated disinfection agent** |
| | (Meine Kommilitonen wählen das notwendige Desinfektionsmittel für eine hygienische Händedesinfektion korrekt aus) |
| 19 | **Fellow students correct me readily, if they perceive a mistake in hand hygiene** |
| | (Meine Kommilitonen korrigieren mich situationsgerecht, wenn sie einen Fehler in meiner Händedesinfektion bemerken) |
| 20 | **Fellow students accept feedback according to the situation if they are corrected by others** |
| | (Meine Kommilitonen nehmen situationsgerecht Rückmeldungen und Hinweise an, wenn sie auf einen Hygienefehler angesprochen werden) |
| 21 | **Physicians conduct hand hygiene if indicated in a situation** |
| | (Ärztinnen und Ärzte führen die hygienische Händedesinfektion situationsgerecht aus) |
| 22 | **Physicians correct me readily, if they perceive a mistake in hand hygiene** |
| | (Ärztinnen und Ärzte korrigieren mich situationsgerecht, wenn sie einen Fehler in meiner Händedesinfektion bemerken) |
| 23 | **Physicians accept feedback according to the situation if they are corrected by others** |
| | (Ärztinnen und Ärzte nehmen situationsgerecht Rückmeldungen und Hinweise an, wenn sie auf einen Hygienefehler angesprochen werden.) |
| 24 | **Nurses conduct hand hygiene if indicated in a situation** |
| | (Pflegekräfte führen die hygienische Händedesinfektion situationsgerecht aus) |
| 25 | **Nurses correct me readily, if they perceive a mistake in hand hygiene** |
| | (Pflegekräfte korrigieren mich situationsgerecht, wenn sie einen Fehler in meiner Händedesinfektion bemerken) |

(*Continued*)

**Table 1.** (Continued)

| No. | Variable |
|-----|----------|
| 26 | **Nurses accept feedback according to the situation if they are corrected by others** |
| | (Pflegekräfte nehmen situationsgerecht Rückmeldungen und Hinweise an, wenn sie auf einen Hygienefehler angesprochen werden) |
| 27 | **The credible maximum effect of omitted hand hygiene is** |
| | **without consequence (1), minor-without any longlasting effect (2), severe- with longer hospital stay (3), critical -with longlasting effect (4), lethal (5)** |
| | (Der glaubwürdig maximale Schaden einer unterlassenen Händedesinfektion ist) |
| 28 | **How often is it that a patient gets harmed in your educational environment?** |
| | **Uncommon (1x >3 years) (1), seldom (once per 3 years) (2), moderate (once per year)(3), often (once per 3 months) (4), very often (once per month) (5)** |
| | (Wie häufig kommt es in Ihrem Ausbildungs—Umfeld vor, dass ein Patient den geschätzten Schaden durch eine ausgebliebene Händedesinfektion erleidet?) |
| 29 | **In the following situations I conduct hygienic hand disinfection in 99% of cases(1), in 75–99% of cases(2), in 25–75% of cases(3), in less than 25% of cases(4), never (5), no answer (-)** <br>• Before greeting a patient with a handshake (WHO-1) <br>• Before positioning a patient on the OP-table (WHO-1) <br>• Before connecting an infusion to an iv-line (WHO-2) <br>• Before connecting a urinary catheter to a collection bag (WHO 2) <br>• Before entering a patient room (no clear indication) <br>• After contamination of the own hand with urine (WHO-3) <br>• After eating in the cantina (no clear indication) <br>• After picking up a patient's used towel (WHO-5) <br>• After positioning a used patient's bed (WHO-5) <br>• After handshake for saying goodbye (WHO-4) <br>• After helping an patient to stand up (WHO-4) <br>• After preparing a sterile i.v.medication (no indication) <br>(In folgenden Situationen setze ich die hygienische Händedesinfektion um in 99% der Fälle (1), in 75–99% der Fälle (2), in 25–75% der Fälle (3), in weniger als 25% der Fälle (4), niemals oder keine Angabe (6) |
| 30 | **If I perceive an error in hygienic hand disinfection I correct other persons** |
| | (Wenn ich einen Fehler in der hygienischen Händedesinfektion bei anderen Personen bemerke, greife ich korrigierend ein bei) |
| 31 | **Who corrects others?** |
| | (Wer greift bei Fehlern anderer Personen ein?) |
| 32 | **In which semester should training in hygienic hand disinfection start?** |
| | (In welchem Semester sollte die Ausbildung in der hygienischen Händedesinfektion beginnen?) |
| 33 | **How often should hygienic hand disinfection be trained/refreshed?** |
| | (Wie häufig sollte die Ausbildung in der hygienischen Händedesinfektion im Sinne einer "Auffrischung" während des Studiums wiederholt werden) |
| 34 | **Please rate the following patient safety issues in order of importance! (1 = most important, 6 = at least important)** <br>• **Medication safety** <br>• **Infection Control** <br>• **Diagnostic Accurancy and Safety** <br>• **Surveillance of Sepsis** <br>• **Cyber- and Data Security** <br>• **Strategies for prevention of unnecessary hospital admissions** <br>(Bitte ordnen Sie folgende Ausbildungsinhalte der Patientensicherheit nach ihrer Wichtigkeit!) |
| 35 | **Please rate the following learning objectives concerning medical education in fection control in order of importance! (1 = most important, 9 = at least important)** <br>• **Personal Protective Equipment skills** <br>• **Hand Hygiene** <br>• **Isolation Precautions** <br>• **Surface Disinfection** <br>• **Medical Product Cleaning and Disinfection** <br>• **Outbreak Strategies** <br>• **Preanalytics** <br>• **Interpretation of laboratory parameters** <br>• **Therapy of infectious diseases** <br>(Bitte ordnen Sie die Inhalte in der Krankenhaushygiene und Infektionsprävention nach Wichtigkeit für die ärztliche Ausbildung!) |
| 36 | **Do you have suggestions, special experience or any comments on infection control?** |
| | (Haben Sie Verbesserungsvorschläge, spezielle Erfahrungen oder Anmerkungen zu Infektionsprävention und Krankenhaushygiene?) |

The survey was provided by the Enuvo GmbH, Zurich, via an online-line link. IP-addresses were blinded towards the investigators by the provider guaranteeing anonymity according to their code of conduct. The online-link was sent via email to the deans of study and student affairs and the heads of the department of hospital hygiene institutions of accredited 45

German, Swiss and Austrian medical faculties in Germany (n = 38), Switzerland (n = 3) and Austria (n = 4) addressing about 120.000 German speaking students. Numbers of medical students were evaluated according to webpages and university rankings.

Data processing, statistics and graphs were conducted using Microsoft Word, PowerPoint and Excel (Microsoft Corporation, Redmond, USA) and XLSTATs (Addinsoft SARL, New York, USA).

Statistical testing included Kruskal-Wallis-tests, Kendall's-Tau ($\tau$) correlations and Friedman-tests adjusted for multiple testing and effect size (intraindividual Cohen's $D_z$) calculation according to Lakens [25].

Qualitative analysis of the "comment section" was conducted using an inductive phenomenological single coder approach [26, 27]. Assignment of tags, coding and theme identification were conducted by the main investigator, an intensive care, hygiene and antimicrobial stewardship expert, post-graduate student in medical ethics with a masters' degree in medical education and ISO 31000 certification in clinical risk management.

## Results

Anonymized data sets are available using the Dryad data suppository (doi.org/10.5061/dryad.tx95x69vx).

### Study population

Altogether 1042 medical students from 12 universities (1 Austrian and 11 German with 1610–3490 medical students hosting altogether about 28000 students,) participated. Thus, of all students internationally addressed, 23% got access to the survey. Of those exposed, 3.5% answered to the questionnaire. Thus, the study population consists of about 0.8% of medical students at German speaking universities.

All semesters (1st to 12th) were enrolled. A few data sets (n = 13) had to be excluded owing to substantial missing (see below) or even zero answers.

Of all students 673 answered all items 10 to 26 about one's own and perceptions concerning others (see Fig 1). Participants not answering at least items 1 to 15 were excluded from the study.

For subgroup analysis participants were allocated to four subgroups to assess for bias by preceding professional experiences, like apprenticeship in nursery, paramedic service and other medical professions (see Table 2):

1. *completers* (group C; students without apprenticeship in any medical field completing the survey).

2. *completers plus* (Group C+; students with apprenticeship in a medical field, completing the survey).

3. *non-completers* (Group NC; students without apprenticeship in any medical field not completing the survey).

4. *non-completers plus* (Group NC+; students with apprenticeship in a medical field, not completing the survey).

Detailed analysis showed, that students without prior education in a medical profession (C, NC) were significantly younger (p<0.001) than those with prior non-academic apprenticeship in a medical field (C+, NC+). Participants in the non-completer groups (NC and NC+)

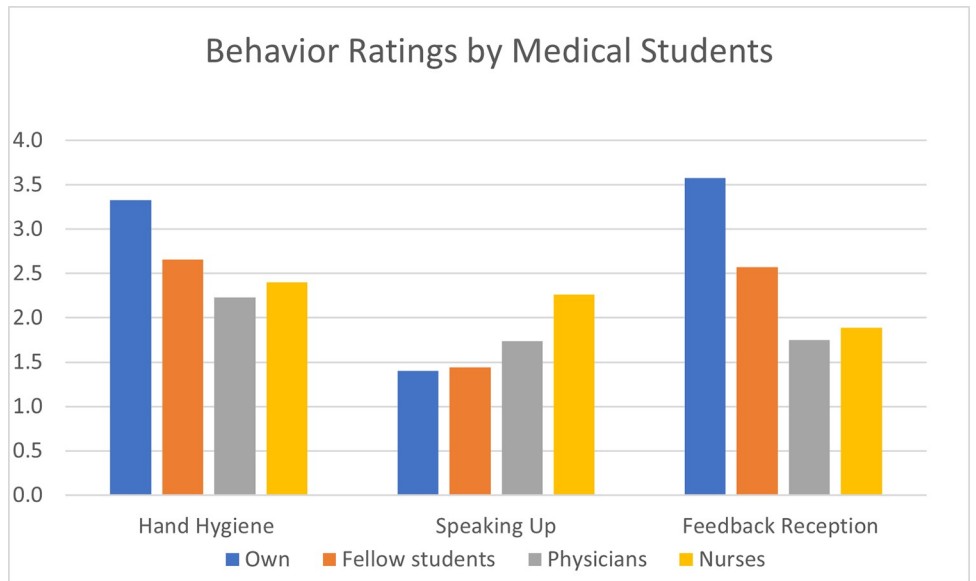

**Fig 1. Ratings by 673 students of all groups for self-assessment (own) and ratings of others (fellow students, physicians and nurses) for the three behaviors as indicated.** Brackets show significances for comparison of own competences (blue bars) with other persons' corrected for multiple testing. In all cases comparing own and others' behavior except for own and fellow students to speak up, data showed significance of p<0.001 corrected for multiple testing.

showed to be exposed earlier to hand hygiene trainings in university (p<0.003). Whereas exposure to the first hand hygiene training occurred at later time in groups C+ and NC+.

## Main results on Hypothesis 1 (existence of overconfidence effects)

**a) Overplacement.** Altogether 673 participants of all subgroups completely answered the items 10 to 26 about their perceptions on one's own and others' skills in hand hygiene and communication. Comparing themselves to others, students rated their adherence to hygiene protocols (mean 3.33) to be significantly better than those of fellow students (2.66), physicians (2.23) and nurses (2.40) with p <0.001 each (see Fig 1 and Table 2). For fellow students, this was reproducible for identification of hand hygiene indications (3.55 vs 2.88, p<0.001). For both comparisons between the participants and fellow students, effect sizes were high (Cohens $D_z > 0.7$).

Concerning omitted or erroneous hand disinfection, students rated to give rarely corrective feedback (mean 1.4 of maximum 4 points). This did not differ from their perception on their fellows correcting them (mean 1.41, p>0.05) and only slightly, but significantly (p<0.001) for estimations of physicians (1.7) and nurses (2.2) correcting them in these occasions.

**Table 2. Demographic data for the complete study population (n = 1042).**

| Group | C | C+ | NC | NC+ |
|---|---|---|---|---|
| Participants (n) | 599 | 169 | 239 | 35 |
| Females (%) | 65.0 | 53.2 | 63.0 | 60.0 |
| Age (years) | 18–35 | 21–37 | 17–34 | 23–35 |
| Mean age | 22.3 | 26.4 | 21.4 | 25.7 |
| Semester (modal value) | 8 | 6 | 5 | 6 |

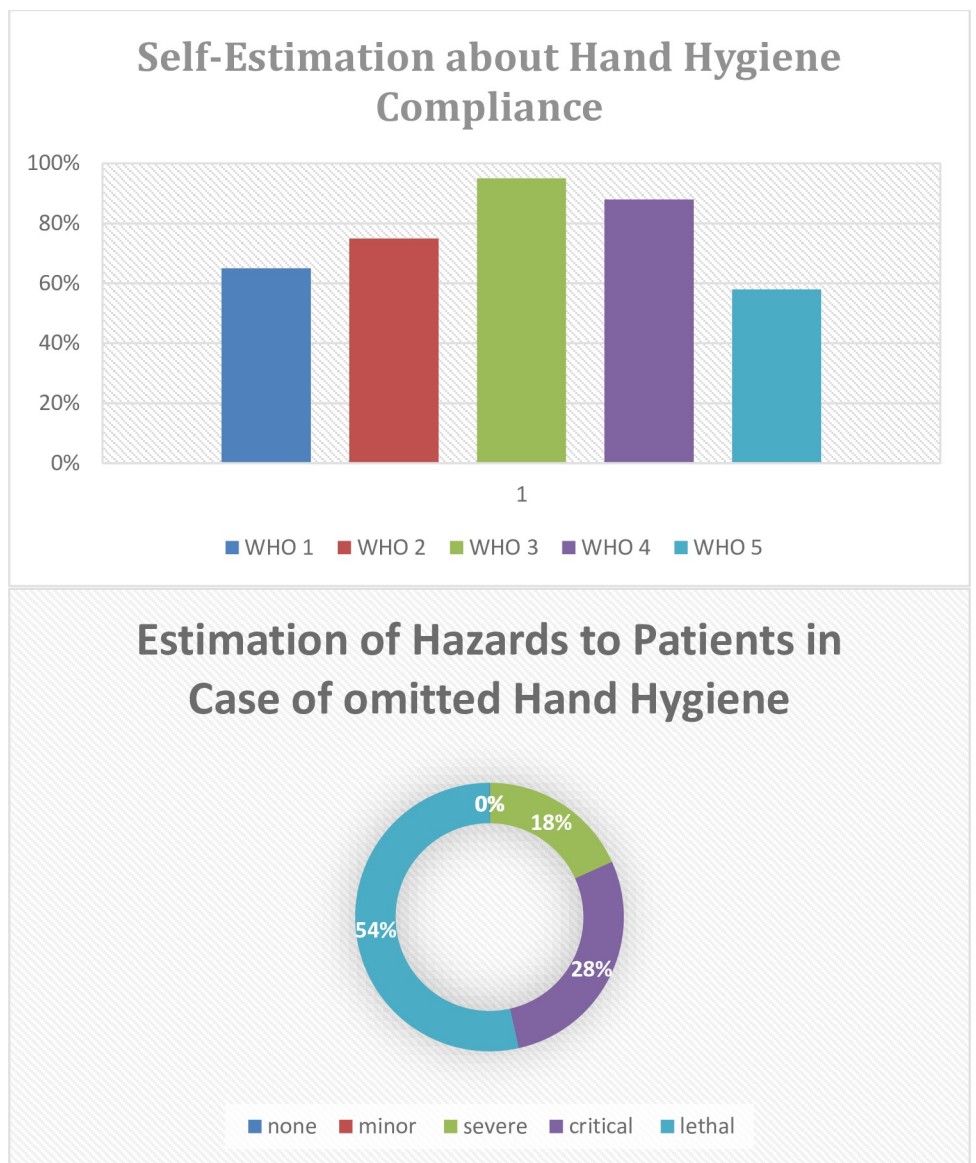

**Fig 2. Ratings by 893 students rating themselves to adhere fairly or completely to hygiene protocols.** (a) Demonstrates the self-estimations in percent of all occasions of compliance to WHO- indications with WHO 1(before touching a patient), WHO 2 (Before aseptic procedures), WHO 3 (after possible contamination), WHO 4 (after touching a patient) and WHO 5 (after contact to the patient's environment). (b) Shows for the same students their reflexing about the maximum possible hazard to a patient in case of an omitted hand disinfection: None (no harm), minor (no sustaining harm, no longer hospital stay), severe (no sustaining harm, but longer hospital stay), critical (life-long impairment) and lethal (death or permanent brain damage).

In contrast, reception-skills for feedback were rated to be very good for own competences (mean 3.5), medium for fellows (2.5) and low for physicians (1.7) and nurses (1.9) with p <0.001 each and very strong effect sizes ($D_z$) of 1.14 for fellows, 1.8 for physicians and 1.67 for nurses.

**b) Overestimation.** Participants of all groups agreeing "fairly" or "completely" to adhere to hand hygiene protocols (n = 893) estimated their own compliance rates to be 65% for WHO—indication 1 (before touching a patient), 75% for WHO–indication 2 (before an aseptic

procedure), 95% after own contamination with body fluids (WHO—indication 3), 88% for WHO indication 4 (after leaving a patient) and 58% for WHO—Indication 5 (after contact to a patient's bed). These students rated the worst credible harm of a missed hand disinfection for patient to be "lethal" (53%), "critical: with long-term impairment (28%), severe: resulting in a longer hospital stay (18%), minor (under 1%) and without any consequence (under 1%). See details in Fig 2.

Likeliness of speaking up was rated to be poor: most students agreed only partially to correct visitors, cleaners and nursery students in case of omitted or faulty hand hygiene (mean 1.9–2.8). Concerning physicians, most rated to completely avoid speaking up on interns (41,1%), consultants (65%) and heads of departments (71%). Only 1.9%, 1% and 0.7% of the participants stated, that they would correct these persons.

## Main results on Hypothesis 2 (dependency of overconfidence on the level of academic education)

Rating own and others' capabilities in hand hygiene and feedback did not correlate with the current semester ($\tau > 0.05$). For some responses concerning self and others, there were very poor significant correlations for age ($\tau < 0.05$, with r ranging from 0.05 to 0.1). Estimations of the maximum credible harm to a patient also did not correlate to age and semester.

Participants of the completer group (most unaffected by preceding medical education, see Fig 3) showed no significant correlation between semester and "overconfidence" (operationalized as the difference between ratings for own and other's competences) with one exception: semester and overconfidence for adherence to hygiene protocols correlated poorly ($p = 0.027$, $\tau = -0.073$). In group C+ similar results could be found with a poor negative correlation for knowledge of indications of hygiene ($p = 0.036$, $\tau = -0.126$).

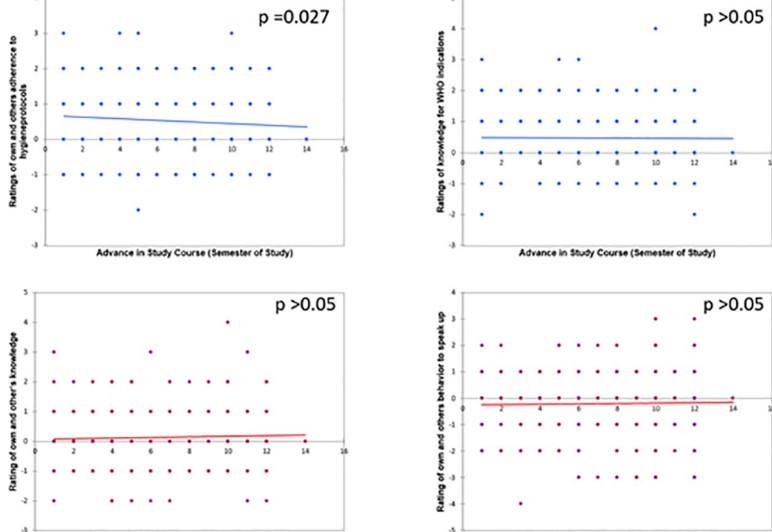

**Fig 3. Correlations of the group C for semester (x-axis) and the mathematical difference between estimation of own and others' competences (y-axis, positive numbers show, that students rated themselves above other, negative that they rated themselves inferior) for adherence to hygiene protocols (upper left, difference of item 10 and 16), knowledge for WHO indications (upper right, items 11 and 17), knowledge about disinfection agents (lower left, items 12 and 18) and the competence to speak up (lower right, items 13 and 19).** Blue lines show negative, red positive correlations.

## Completer/non-completer comparison

For ratings of own and others' competencies and risks of omitted hand hygiene no differences between completers and non-completers[24] could be detected.

## Further results on experienced speaking up behavior

Concerning a matrix question "who corrects whom" (Fig 4) in occasion of errors or omissions in hand hygiene, subjective ratings of the students showed, that highest correctional interventions were conducted by nurses. The group to be corrected least were post-graduate physicians, especially if they were of higher hierarchic level.

## Further results on the perception of the hygiene curricula and impact on patient safety

Most participants (68%) agreed, that the first lessons in hand hygiene should be held in the very first semester and should be repeated once per semester (29.5%) or year (37.8%). Concerning rankings of patient safety competences participants rated infection prevention (1st place) and medication safety (2nd) to be most relevant for their own medical education. Diagnostics (3rd) and treatment of sepsis (4th) were followed by strategies to prevent unnecessary hospital admissions (5th) and cyber security (6th). For their education in hospital hygiene they rated the importance of hand hygiene (1st place) above skills in personal protective equipment (2nd), barrier precautions (3rd), diagnostic (4th) and therapy (5th) of infectious diseases, pre-analytics(6th), outbreak treatment (7th), surface disinfection (8th) and medical product decontamination (9th).

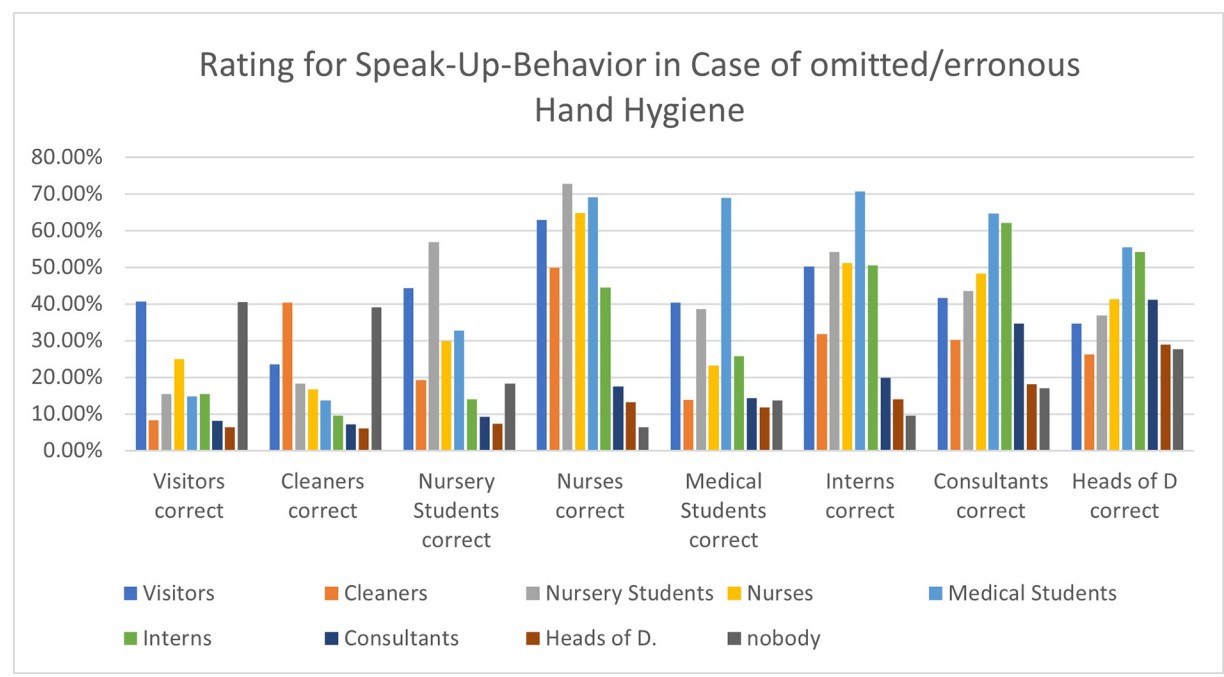

**Fig 4. Correctional behavior perceived by medical students (groups C und C+ groups).** On the Y-axis the percentages of agreements for the question if a specific group (x-Axis) corrects others (colored columns) is shown. Most corrections on others and the own profession are perceived for nurses, while cleaners and visitors are not perceived to do corrections on others. Medical students themselves feel to be corrected by all others groups. Concerning physicians, they rated to be less corrected with rising levels of hierarchy.

## Qualitative data analysis

Altogether 164 free-text entries (word count 31911) were evaluated and 55 codes were assigned in an iterative process. We identified four main subjects:

**1) Present education of physicians and students in hand hygiene is insufficient.** Students reported about methodically weak hygiene lectures ("*boring*", "*hygiene is a dry field*", "*false didactic concepts*", "*better use of emotional motives*", "*there should be relevant tests and trainings on simulations*") and especially, that the first exposure to infection control mainly is located after the first contact to patients. Many participants argued, that the first hygiene training should take place "*before first patient contact in the care traineeship*". A second common message was, that there are profound differences between professions in quality and quantity of hygiene training (nurses, physiotherapists, paramedics). Additionally, many reported about weak attitudes in hygiene culture like "*arbitrariness of rules*" and "*sloppy hygiene*" of physicians linked to age and hierarchical status ("*the older the doc, the sloppier hand hygiene*", "*interns are frowzy with hand hygiene, but I don't even want to speak about consultants*"). Low attitude of fellows and physicians and active ignorance of rules were common tags ("*some doctors think, they are immune to transmit microbes*", "*ignorance*", "*hygiene is a medical stepchild*", "*There is a fellow, who never washes his hands after urination*"), sometimes related to economic reasons and ethical issues ("*hygiene rules and decontamination of operation-theatres are actively sacrificed to stick to the schedule*").

**2) Implementation of feedback-culture is inhibited by hierarchy and clinical tribalism.** Many students reported about cultural barriers concerning feedback. "*Pecking order*", "*nobody dares to correct the boss*", "*hygiene would profit a lot, if the structures in hospitals would not be defined by might and hierarchy*" and "*physicians do not want feedback*" were common answers. Additionally, clinical tribalism effect [28] (a delimitation of professionals: "we against them") could be detected, especially in experienced students (C+): "*during my work as a nurse [. . .] I experienced, that doctors did not want me to show them how to catheterize a patient*", "*I have difficulties to speak up on nurses. Mainly, they are not speaking highly of us. [. . .] It's not easy for them to be corrected by us*", "*I my first job so far, I did not experience colleagues to be 'pissed-off' if hygiene rules were explained. [. . .] This I experienced first in med school.*"

**3) Hygiene and feedback are issues of medical ethics.** Some students reported about ethical issues linked to hand-hygiene. One person suggested to implement Aristotelian ethics ("*every passionate doctor can be enthused for his specialty and helps his patient.*") or Kantian (deontological) approaches (categorical imperative) to education and feedback ("*What would you do, if your children, partner, parents or grandparents would* [be affected]").

**4) There is no consequence for breaking hygiene rules.** A lot of comments, especially if students reported about non-adherence to hygiene rules were marked with exclamations marks, coded by us as signs of emotional reactions. Further, emotions like "fear" and anger and a feeling of frustration and helplessness were reported ("*Critical thinking and criticism and to open your eyes and your mouth are not taught in systems teaching for learning of details and tests. But you do, what you are expected to do. Sadly.*"). Further they reported about missing consequences in terms of incompliance to hygiene rules ("*There is no consequence, if hygiene rules are broken. This is why they are violated.*") and the need for more regularly controls ("*Surprising visits of hygiene experts*" and "*experts disguised as students*"), control systems ("*Control system with penalties*") and education tests. Last but not least, the discrepancy of hygiene rules and real behavior of role models, and the impact on medical education was a very common answer ("*The boss should respect the hygiene rules*", "*you cannot expect students to put something into practice, they never got demonstrated*", "*you are learning hygienic behavior from idols, not lectures*").

## Discussion

To our knowledge this is the first study in medical students combining assessments of own and others' competencies for hand hygiene and speaking-up competences.

### Main results and hypotheses

Under consideration of the selection bias (see below) hypothesis One (presence of overconfidence) could be confirmed for hand hygiene and feedback reception, but not for feedback provision. Participants rated themselves significantly superior to fellow students in hand hygiene and feedback reception competences (overplacement), but not in Speaking-Up behavior. As overconfidence occurs mainly in abilities perceived to be easy [15], this may indicate that students experience hand hygiene to be of low, but speaking up of high difficulty. The result of rating feedback provision skills not above others competences is contrary to our study in post-graduates [13], where overplacement could be shown. An explanation for this could be, that under-graduates experience barriers and helplessness in speaking up more often than post-graduates do, who correct others more often.

Only 54% of the participants rating themselves to be good or very good trained in hygiene realized the potentially lethal relevance of an omitted hand disinfection. This indicates either for overestimation [17] (with de facto missing knowledge) for the whole competency, or relevant deficits in attitude. Nevertheless, these results confirm prior findings of our working groups in a regional [12] and national [13] post-graduate settings and further results in medical [20–22] and non-medical [14, 15, 29] context.

The second hypothesis (education-dependence of overconfidence) could not be confirmed. Ratings were independent from level of education, especially in the "naïve" group C. For this study it was our motivation, to identify the "onset" of overconfidence for an early curricular intervention. Consequently, it seems to be reasonable to say that overconfidence develops after the first exposure to the proficiency [15]. In Germany, medical students have to complete a hospital traineeship before admission to university or within the first four semesters. It is likely, that overconfidence is acquired in this time prior to regular patient contact. Consequently, a governmental structured pre-academic traineeship using reflective methods, known to potentially lower overconfidence [29, 30], and focusing on hand hygiene and feedback may be indicated.

We did not find any other correlations of high reliability ($\tau > 0.6$) which is consistent with prior general findings [15, 17, 31] and results of our research in medical post-graduates [13]. With overconfidence being an ubiquitarian effect independent from academic progress, a change for curricular response to compensate the effect [32] may not be of satisfactory effect. Instead, further research could concentrate on the first exposure to hand hygiene in clinical settings, which may be the cause for the overconfidence, if the attitude learning dimension [33] is not addressed properly.

### Speaking-up behavior and patient safety

Our results show an overall low readiness to speak up, even towards persons hierarchically subordinated in clinical settings. Additionally, medical students face intense barriers to speak up on physicians of all hierarchies. These findings are consistent with a study by Schwappach et al. [34] and others [35, 36]: Hierarchic structures with fear for negative consequences or being wrong in case of given feedback were mentioned several times and seem to more powerful than the ethical codex to protect patients from any potential harm. Thus, even two decades after the groundbreaking publication "*To Err is human*" [37], a constructive human factor

management has not progressed to the desirable level. As literature shows, this is not limited to German work places [38] or infection prevention [39, 40] and is moreover a general effect.

On the other hand, students mentioned the needs for change in human factor management several times in the comments section. Conclusively, medical education of pre- and postgraduates is a desired and demanded key element to change mindsets towards a constructive working culture and team interaction. To do so, medical staff of all educational levels have to develop and maintain feedback strategies [41–43] and the attitude and motivation to use them.

## Qualitative results

In this study, four main themes could be identified. The first theme (insufficiency of medical training in infection prevention) showed high resistances of students and medical staff towards hygiene trainings. These were mentioned to be "*boring*", "*annoying*" and "*dry*" which is consistent with finding from 1994 [44]. Those could be overcome by more attractive and "gameful" formats [45, 46]. Second, hierarchical barriers and clinical tribalism ("*we against them*") reduce the possibility of feedback and intervention, as discussed above. The third main theme, shows demand for ongoing research towards ethical considerations and of dilemmatic factors leading to a potentially "social maladaptation" to actively accept impairment of patient safety due to hierarchical issues. The fourth mentioned, that students get frustrated as they feel not to be part of the staff and that feedback is not of relevance if someone speaks up. With medical students known to be distressed due to intensive psychosocial pressure and even burn-out [47] our findings suggest, that they have to weigh, if hierarchical unwelcome speaking up is too resource-consuming and thus further frustrating.

Conclusively, the combination of students being overconfident for hygiene skills, the experience that non-adherence to hygiene protocols is without consequence, speaking-up is potentially linked to punishment and frustration, it is likely that motivation to visit hygiene trainings is reduced, especially if these are "dry", "boring" and "far away from reality". Thus, our initial question has to be changed to *"Why should I go to a time-consuming and boring training for methods and rules I already mastered and which are not followed by my teachers themselves?"*

## Results of students' educational demands

As the results suggests, under-graduates themselves recommended early training in hygiene and feedback, before first contact to patient or skills with demand for hygiene management. Further evaluation showed, that participants see high demand for infection prevention and medication safety skills. For hand hygiene, participants clearly stated that this skill should be trained annually from the very beginning. Additionally, the results depict high demand for education in use of personal protective equipment and management of multi-drug resistant organisms.

## Limitations

About 0.8% of the German-speaking medical students participated in our survey [48] leading to the question of representativeness. It seems plausible, that those interested in hygiene, participated in this study and may be de facto better than those who did not. This is known as the self-selection recruitment and non-response bias [49] limiting most open survey studies.

To compensate, we used the completer/non-completer approach [24], a comparison to our preexisting studies [12, 13] and we investigated external literature [17, 29]. Following findings and arguments lower the selection bias (without ruling in out) and the counter-hypothesis, that motivated participants are de facto better in their competences:

1. The completer/non-completer approach showed no difference between those motivated to complete the survey and those not. If interest in the topic would be an argument for competence, those not interested (non-completers), should be of lesser performance in the tests for overestimation and should show less overconfidence. This was not the case.

2. Motivation to contribute to surveys is not depending on interest in the subject only, but also on politeness and endorsement by providers [24]. We do not know that for this investigation.

3. A post-hoc analysis of the participants response rate in groups "C" showed a return rate for contributing universities between 0.12% and 4.1% of the addressed students in these institutions. This would mean, that in those universities with low response rates, students would be less motivated for hygiene trainings. According to the counter-hypothesis they should be worse, also in identification of risks. For items representing the knowledge concerning risks for omitted hand hygiene (item 27 and 28) and self-estimation for compliance (items 29 and 30), we did not detect any difference between students of different universities.

However, these clues do not break the potential bias [50], but limit it. Nevertheless, this study is a multicenter trial showing similar effects in different universities despite assumable different distribution and endorsements of the study and our results are in concordance with psychological investigations from different authors and in different populations.

But, even if participating "biased" students were among the "best" and "most interested", our results of only 53% of them understanding the potentially lethal effects of omitted hand hygiene, most not speaking up, as well as their statements concerning hygiene education are alarming.

A minor limitation is, that we did not ask for the curricular methods of medical education in hygiene and communication skills differing between the universities. Thus, it may be possible, that there were students from a university less overconfident, but better educated leading to the same net results. Therefore, further work should evaluate the different methods of the universities involved to screen for a hidden effect of the curricula.

Third, we asked about omitted hand hygiene occasions, but not about the cause for it. For further studies, a differentiation between slips and lapses, mistakes and active violations should be considered [51].

## Conclusion

Our results confirm the hypothesis that overconfidence effects in hygiene and feedback are present in medical students as they are in postgraduates. These effects are independent from age, gender, semester and prior education. For patient safety, students reported about hierarchical barriers, fear of punishment and absence of consequences for broken hygiene rules leading to frustration and disruption of theory and practice in infection prevention. For education, our study showed the suggestions to implement creative and appealing hygiene programs early in education with annually repetition paired with ongoing reduction of hierarchical barriers. However, our findings are limited by the selection bias especially concerning self-recruitment, so that further work should concentrate on this limitation. Altogether, all health care providers face the demand for change of mindsets concerning hand hygiene and speaking-up practice to develop a constructive work place culture and error management.

## Supporting information

**S1 Data.**
(XLSX)

**S1 File. German version of the questionnaire.**
(PDF)

## Acknowledgments

We thank all contributing universities and students for participating.

## Author Contributions

**Conceptualization:** Stefan Bushuven, Sonia Sippel, Sarah Koenig.

**Data curation:** Stefan Bushuven, Stefanie Bushuven.

**Investigation:** Stefan Bushuven.

**Methodology:** Stefan Bushuven.

**Project administration:** Stefan Bushuven.

**Software:** Stefan Bushuven.

**Supervision:** Markus Dettenkofer.

**Validation:** Markus Dettenkofer, Sarah Koenig, Wulf Schneider-Brachert.

**Visualization:** Stefan Bushuven.

**Writing – original draft:** Stefan Bushuven.

**Writing – review & editing:** Sarah Koenig, Wulf Schneider-Brachert.

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
