## [Decision Letter · Decision Letter 0]

22 Apr 2020

PONE-D-20-03135

Speaking Up Behavior and Cognitive Bias in Infection Control: Competences of German Medical Students

PLOS ONE

Dear Dr. Bushuven,

Thank you for submitting your manuscript to PLOS ONE. After careful consideration, we feel that it has merit but does not fully meet PLOS ONE’s publication criteria as it currently stands. Therefore, we invite you to submit a revised version of the manuscript that addresses the points raised during the review process.

We would appreciate receiving your revised manuscript by Jun 06 2020 11:59PM. To enhance the reproducibility of your results, we recommend that if applicable you deposit your laboratory protocols in protocols.io, where a protocol can be assigned its own identifier (DOI) such that it can be cited independently in the future. For instructions see: http://journals.plos.org/plosone/s/submission-guidelines#loc-laboratory-protocols

We look forward to receiving your revised manuscript.

Kind regards,

Denis Bourgeois

Academic Editor

PLOS ONE

Journal Requirements:

3. Please include your tables as part of your main manuscript and remove the individual files. Please note that supplementary tables (should remain/ be uploaded) as separate "supporting information" files

Additional Editor Comments (if provided):

Reviewers' comments:

Reviewer's Responses to Questions

**Comments to the Author**

1. Is the manuscript technically sound, and do the data support the conclusions?

Reviewer #1: Yes

Reviewer #2: Yes

2. Has the statistical analysis been performed appropriately and rigorously? 

Reviewer #1: Yes

Reviewer #2: I Don't Know

3. Have the authors made all data underlying the findings in their manuscript fully available?

Reviewer #1: No

Reviewer #2: No

4. Is the manuscript presented in an intelligible fashion and written in standard English?

Reviewer #1: Yes

Reviewer #2: Yes

5. Review Comments to the Author

Reviewer #1: This study focuses on the speaking up behavior and hand hygiene skills of german-speaking medical students.

The manuscript is technically sound and the topic of importance, since infection control procedures are the cornerstone for quality care. Tracks are given as to the time of hygiene education during the curricula. A focus is made on the relations between students and their hierarchy, suggesting a constructive questioning of the practices of seniors.

The authors have not made all data underlying the findings in their manuscript fully available directly in their manuscript, but they have specified that “the data can be shared in German language and after blinding of text passages referring directly to the universities contributing”.

In general, authors should avoid the coma before the word “that”, probably due do the German “dass” which usually follows a coma.

Title: infection control is a wide field. The title should reflect the study, replacing “infection control” by “hand hygiene”. Moreover, the study dealt with “german-speaking medical students”, and not only “german medical students”.

Abstract: abstract should be reread and minor English mistakes should be corrected (“students ask to participated”, “date” for “data” …).

“All universities were contacted and students…”, please precise “all medical students”.

Introduction: the end of the paragraph about overconfidence is unclear. Last sentence: do you talk about causes for these effects, or consequences for these effects? If causes, it would be welcome to explain in 2-3 sentences how these causes interact with the overconfidence effect.

Material and methods: what online tool has been used to collect answers? How did that tool guarantee the anonymity of the participants? Who were the 51 persons who previously tested the questionnaire?

Items 1-9, what kind of demographic data did you collect? Please specify.

Results: 13 data sets were excluded because of missing answers. From how many missing responses was the participant still included in the study, since 2 groups contained non-completers? This point is unclear, and the definition of a non-completer should be stated. There are extra- or missing-parenthesis in the description of the 4 groups.

In the overestimation paragraph, comas after “critical” and “severe” should be replaced by colons.

In table 2, why have you chosen not to give the mean age ?

In figure 2, “a” and “b” are missing.

Discussion: the sentence “Consequently, it seems to be reasonable, that overconfidence develops after the first exposure to the proficiency” sounds wrong. I think it should be “Consequently, it seems reasonable to say that overconfidence develops after…”

In the “qualitative results” section, and regarding the first theme, what about mentioning works that have been carried to attract students to hygiene eeducation? Pedagogical games can be mentioned as interesting ways. For example: Morrell BLM, Ball HM. Can You Escape Nursing School?: Educational Escape Room in Nursing Education. Nurs Educ Perspect. 2019 Jan 3. doi: 10.1097/01

or original methods that could be implemented, for example: Offner D, Strub M, Rebert C, Musset AM. Evaluation of an ethical method aimed at improving hygiene rules compliance in dental practice. Am J Infect Control. 2016 Jun 1;44(6):666-70. doi: 10.1016/j.ajic.2015.12.040

Reviewer #2: This research try to determine how medical students experience their difficulties (e.g overconfidence, speaking-up,..) for hand hygiene in daily routine

The introduction provide sufficient background and include all relevant references

The research problem is well presented and the methods seem to be adequate. The results are interesting and discussion is consistent. I do have some minor comments which the authors might consider.

In the title : We suggest to replace « German Medical Students » by « German speaking medical students » since you indicate that the study include also German and Austrian students

In Material and Method :

-we suggest to move (see table 1) after Items 10-26 since table 1 doesn’t contain 36 items

“The survey consisted of 36 items. Items 1-9 evaluated demographic data and prior apprenticeships in a medical profession. Items 10-26 (see table 1) assessed the subjective personal”

In Results :

-If we well understand, Swiss faculties did not answer, so you should remove Switzerland from the results paragraph

“Thus, the study population consists of about 0.8 % of medical students in Germany, Austria and Switzerland at German speaking universities.”

- In the text, we read “These students rated the worst credible harm of a missed hand disinfection for patient to be lethal (53%)…” whereas the figure 2c shows 54%

Personally, I think that this figure 2 is less useful than the others

In discussion :

I suggest to make reference to studies such as :

Offner D, Strub M, Rebert C, Musset AM. Evaluation of an ethical method aimed at improving hygiene rules compliance in dental practice. Am J Infect Control. 2016;44(6):666–670. doi:10.1016/j.ajic.2015.12.040

Finally, I would like to say that this type of study is very pertinent not only because it’s a hot topic at the present time but the rules of hygiene and asepsis are very important at any time in preventive and curative health care.

6. PLOS authors have the option to publish the peer review history of their article (what does this mean?). If published, this will include your full peer review and any attached files.

Reviewer #1: No

Reviewer #2: No

---

## [Author Response · Author response to Decision Letter 0]

22 Aug 2020

We thank both reviewers for valuable information and suggestions about our manuscript. I hope our changes and corrections meet the expectations for publication.

---

## [Editor Report · Decision Letter 1]

8 Sep 2020

Speaking Up Behavior and Cognitive Bias in Hand hygiene: Competences of German-speaking Medical Students

PONE-D-20-03135R1

Dear Dr. Bushuven,

We’re pleased to inform you that your manuscript has been judged scientifically suitable for publication and will be formally accepted for publication once it meets all outstanding technical requirements.

Kind regards,

Denis Bourgeois

Academic Editor

PLOS ONE
---

## [Editor Report · Acceptance letter]

16 Sep 2020

PONE-D-20-03135R1 

*Speaking Up Behavior and Cognitive Bias in Hand Hygiene: Competences of German-speaking Medical Students*

Dear Dr. Bushuven:

I'm pleased to inform you that your manuscript has been deemed suitable for publication in PLOS ONE. Congratulations! Your manuscript is now with our production department. 

Kind regards, 

on behalf of

Professor Denis Bourgeois 

Academic Editor

PLOS ONE